# RNA transcripts serve as a template for double-strand break repair in human cells

Manisha Jalan [1,13], Alessandra Brambati[2,13], Hina Shah [2,3,13], Niamh McDermott[1,13], Juber Patel[1], Yingjie Zhu [4], Ahmet Doymaz[2,3], Julius Wu[2,5], Kyrie S. Anderson[1], Andrea Gazzo[4], Fresia Pareja [4], Takafumi N. Yamaguchi[6,7,8], Theodore Vougiouklakis[4], Sana Ahmed-Seghir[1], Philippa Steinberg[6,7], Anna Neiman-Golden[6,7], Benura Azeroglu[9], Joan Gomez-Aguilar[1], Edaise M. da Silva [4], Suleman Hussain[1], Daniel Higginson [1], Paul C. Boutros [6,7,8,10,11], Nadeem Riaz [1], Jorge S. Reis-Filho[4,12], Simon N. Powell [1,2] ✉ & Agnel Sfeir [2] ✉

Double-strand breaks (DSBs) are toxic lesions that lead to genome instability. While canonical DSB repair pathways typically operate independently of RNA, growing evidence suggests that RNA:DNA hybrids and nearby transcripts can influence repair outcomes. However, whether transcript RNA can directly serve as a template for DSB repair in human cells remains unclear. In this study, we develop fluorescence and sequencing-based assays to show that RNA-containing oligonucleotides and messenger RNA can serve as templates during DSB repair. We conduct a CRISPR/Cas9-based genetic screen to identify factors that promote RNA-templated DSB repair (RT-DSBR). Of the candidate polymerases, we identify DNA polymerase zeta (Polζ) as a potential reverse transcriptase that facilitates RT-DSBR. Furthermore, analysis of cancer genome sequencing data reveals whole intron deletions - a distinct genomic signature of RT-DSBR that occurs when spliced mRNA guides repair. Altogether, our findings highlight RT-DSBR as an alternative pathway for repairing DSBs in transcribed genes, with potential mutagenic consequences.

The human genome is constantly exposed to endogenous and exogenous insults that cause DNA damage. Among the various types of DNA damage, double-strand breaks (DSBs) are particularly harmful, leading to genome instability, a hallmark of aging, cancer, and neurodegeneration[1]. DSBs are repaired through three major pathways: homologous recombination (HR), non-homologous end-joining (NHEJ), and microhomology-mediated end-joining (MMEJ). NHEJ repairs DSBs by ligating broken ends with minimal processing. HR and MMEJ rely on DNA resection to generate a single-stranded DNA tail that anneals to the sister chromatid or the opposing DSB end[2]. While these canonical repair pathways generally function independently of RNA, ~78% of the genome is actively transcribed at any given time[3]. As a

[1]Department of Radiation Oncology, Memorial Sloan Kettering Cancer Center, New York, NY, USA. [2]Molecular Biology Program, Sloan Kettering Institute, Memorial Sloan Kettering Cancer Center, New York, NY, USA. [3]Weill Cornell Medicine, Cornell University, New York, NY, USA. [4]Department of Pathology and Laboratory Medicine, Memorial Sloan Kettering Cancer Center, New York, NY, USA. [5]SUNY Downstate Health Sciences University, New York, NY, USA. [6]Department of Human Genetics, University of California, Los Angeles, CA, USA. [7]Jonsson Comprehensive Cancer Centre, University of California, Los Angeles, CA, USA. [8]Institute for Precision Health, University of California, Los Angeles, CA, USA. [9]Laboratory of Genome Integrity, National Cancer Institute (NCI), National Institutes of Health (NIH), Bethesda, MD, USA. [10]Department of Urology, University of California, Los Angeles, CA, USA. [11]Broad Stem Cell Research Center, University of California, Los Angeles, CA, USA. [12]Present address: AstraZeneca, Gaithersburg, MD, USA. [13]These authors contributed equally: Manisha Jalan, Alessandra Brambati, Hina Shah, Niamh McDermott. ✉e-mail: powells@mskcc.org; sfeira@mskcc.org

result, DSB repair frequently occurs in open chromatin regions, and RNA transcription must be intricately coordinated with DNA repair to ensure genomic stability and appropriate gene expression.

Over the years, research into the interplay between transcription and DSB repair uncovered how DSBs modulate gene expression and how transcription, in turn, shapes repair outcomes. While the activation of DNA damage signaling kinases, ATM and DNA-PK, was shown to repress transcription by RNA Pol II near break sites[4–6], conflicting results suggested that enhanced transcription generates noncoding RNAs that amplify DNA damage signaling and recruit HR factors to DSB sites[7–10]. RNA transcripts accumulating at break sites can anneal to DNA, leading to the formation of RNA:DNA hybrids[11–15] that promote DSB repair by regulating DNA end-resection and facilitating the recruitment of repair factors[16]. RNA transcripts have also been shown to stimulate HR by invading the donor DNA in response to DSBs, forming an intermediate D-loop containing RNA, which increases the accessibility of the break to the donor DNA template[17].

Beyond its indirect role in orchestrating DSB repair, RNA can also play a more direct, instructive role by serving as a template for DSB repair. In *Saccharomyces cerevisiae*, it has been demonstrated that messenger RNA (mRNA) can be reverse transcribed to act as a template for DSB repair in contexts where RNaseH1 and RNaseH2 are lacking. One form of RNA templated DSR Repair (RT-DSBR) involves the production of a cDNA intermediate by Ty retrotransposons, which is then used as a template for HR repair. Another mechanism entails base pairing of the mRNA with single-stranded DNA (ssDNA) flanking the break site, followed by its copying in cis by the translesion polymerase zeta (Polζ)[18–20]. Whether RT-DSBR is conserved in higher eukaryotes remains unknown.

In human cells, the transfer of genetic information from RNA to DNA is predominantly mediated by two reverse transcriptase activities. The first is telomerase, which reverse transcribes its RNA to replenish telomere DNA[21]. The second reverse transcriptase activity involves ORF2, which reverse transcribes LINE-1 RNA into DNA, allowing the integration of the transposable element into the genome[22]. Both ORF2 and telomerase activities have been detected at DSBs induced by CRISPR/Cas9 cleavage, where LINE-1 and TTAGGG are introduced, albeit with very low efficiency[23,24]. In addition, biochemical studies have shown that while several human replicative and translesion polymerases can copy up to 2 and 3 embedded ribonucleotides, Polymerase theta (Polθ-encoded by *POLQ*) can reverse transcribe several kilobases of mRNA in vitro[25]. This reverse transcriptase activity has been suggested to facilitate RNA-templated repair in vivo. However, whether mRNA can act as a template for DSB repair in human cells and which enzyme is responsible for mediating this reverse transcription remains uncertain.

In this study, we explore the direct role of RNA templating during DSB repair in human cells by developing complementary fluorescence- and sequencing-based reporter assays. Our findings reveal that both RNA-containing oligonucleotides and endogenous RNA transcripts can act as donor templates for RT-DSBR in human cells. Through a CRISPR/Cas9 screen, we identify and validated the translesion polymerase Polζ as the reverse transcriptase that promotes RT-DSBR. We hypothesize that when spliced mRNA is used as a template, RT-DSBR would result in the deletion of introns. To test this, we examine the repair of a DSB occurring within introns using spliced transcripts as templates and find that this leads to complete intron removal from the genome. Leveraging the phenomenon of intron loss, we provide evidence of RT-DSBR under physiological conditions. By analyzing sequencing data from MSK-IMPACT (Memorial Sloan Kettering-Integrated Mutation Profiling of Actionable Cancer Targets)[26,27] and Pan-Cancer Analysis of Whole Genomes (PCAWG)[28], we identify precise deletions of intronic sequences, which we refer to as whole intron deletions (WIDs). These WIDs serve as genomic signatures of RT-DSBR activity in cancer genomes, supporting the idea that spliced mRNA can

serve as a repair template for DSBs. Collectively, our findings suggest that RNA can function as a repair template for DSBs. We propose that RT-DSBR plays a particularly significant role in regions of high transcriptional activity, offering an additional mechanism for maintaining genome integrity.

## Results

### Human cells repair DSBs using RNA as a template to copy genetic information

To investigate whether RNA can directly serve as a template during DSB repair in human cells, we developed two complementary reporter assays capable of detecting reverse transcription activity at a CRISPR/Cas9-induced DSB using either fluorescence or sequencing readouts. In the first assay, we used a blue fluorescent protein (BFP)-to-green fluorescent protein (GFP) reporter system that involves a single amino acid change (His66Tyr) detectable by flow cytometry (Fig. 1a and Supplementary Fig. 1a, b)[29]. As previously demonstrated, repairing a CRISPR/Cas9-induced DSB in a BFP gene, randomly integrated into the genome, successfully converted BFP to GFP. This conversion occurred when a single-stranded DNA donor (DNA$^{GFP}$) containing the corresponding amino acid change was used as the repair template[30]. To adapt this reporter for RNA-templated DSB Repair (RT-DSBR), we generated chimeric oligonucleotide donors by replacing the three bases coding for tyrosine in the DNA template with the corresponding ribonucleotides (rNTPs), creating a series of donors with 3 to 15 ribonucleotides spanning the His66Tyr sequence (DNA/RNA$^{3R/6R/8R/15R}$) (Fig. 1b). The successful templated repair of BFP, resulting in GFP expression, is anticipated to be mediated by a reverse transcriptase that copies the donor's RNA residues into DNA.

Using HEK293T cells stably expressing BFP, we delivered Cas9 protein and sgRNA targeting BFP, along with the DNA or DNA/RNA chimera donors. In the presence of the DNA$^{GFP}$ donor, approximately 70% of cells exhibited gene disruption without templated repair (GFP$^-$BFP$^-$), whereas 25% of cells showed BFP-to-GFP conversion (GFP$^+$; Fig. 1b). Repair efficiency was quantified by fluorescence-activated cell sorting (FACS), and the codon-switch was validated by sanger sequencing (Fig. 1a and Supplementary Fig. 1a–c). Repair of Cas9-induced breaks using DNA/RNA chimeric donors resulted in lower, albeit still significant, percentages of GFP-positive cells, implicating a reverse transcriptase activity that could synthesize up to 15 rNTPs during DSB repair (Fig. 1b). A chimera donor with three scrambled rNTPs did not result in any significant increase in GFP$^+$ cells, ruling out random mutagenesis associated with CRISPR/Cas9 editing (Supplementary Fig. 1d). Additionally, RNaseA treatment and gel electrophoresis demonstrated that DNA contamination did not interfere with the assay (Supplementary Fig. 1e).

In a complementary approach, we induced a Cas9 break at the safe harbor genomic *AAVS1* locus and provided donor oligos with a unique three base-pair (bp) insertion (GAT) (Fig. 1c)[31]. The donor was either a pure ssDNA (DNA$^I$) or a DNA/RNA chimera donor containing 10 or 22 ribonucleotides spanning the three bp insertion (DNA/RNA$^{10R}$ or DNA/RNA$^{22R}$) (Fig. 1d and Supplementary Data 1.2). We assessed repair frequency via next-generation sequencing (NGS) on a 245 bp amplicon flanking the Cas9 cut site. Using CRISPResso2[32], we identified and quantified the RNA insertion as a fraction of the total insertions and deletions (indels). The AAVS1-seq assay detected repair events in the presence of a DNA/RNA chimera, thus corroborating data obtained from the BFP-to-GFP conversion assay (Fig. 1d). We further validated these results with droplet digital PCR (ddPCR) using probes to detect insertions at the break site (Supplementary Fig. 1f, g and Supplementary Data 1.1). We observed a strand bias with DNA/RNA donors compared to ssDNA oligos (Supplementary Fig. 1h–i), similar to previous observations for single-strand templated repair (SSTR) at Cas9-induced breaks[33], indicating that RNA-containing donors are directly copied at the break site without requiring a double-stranded DNA

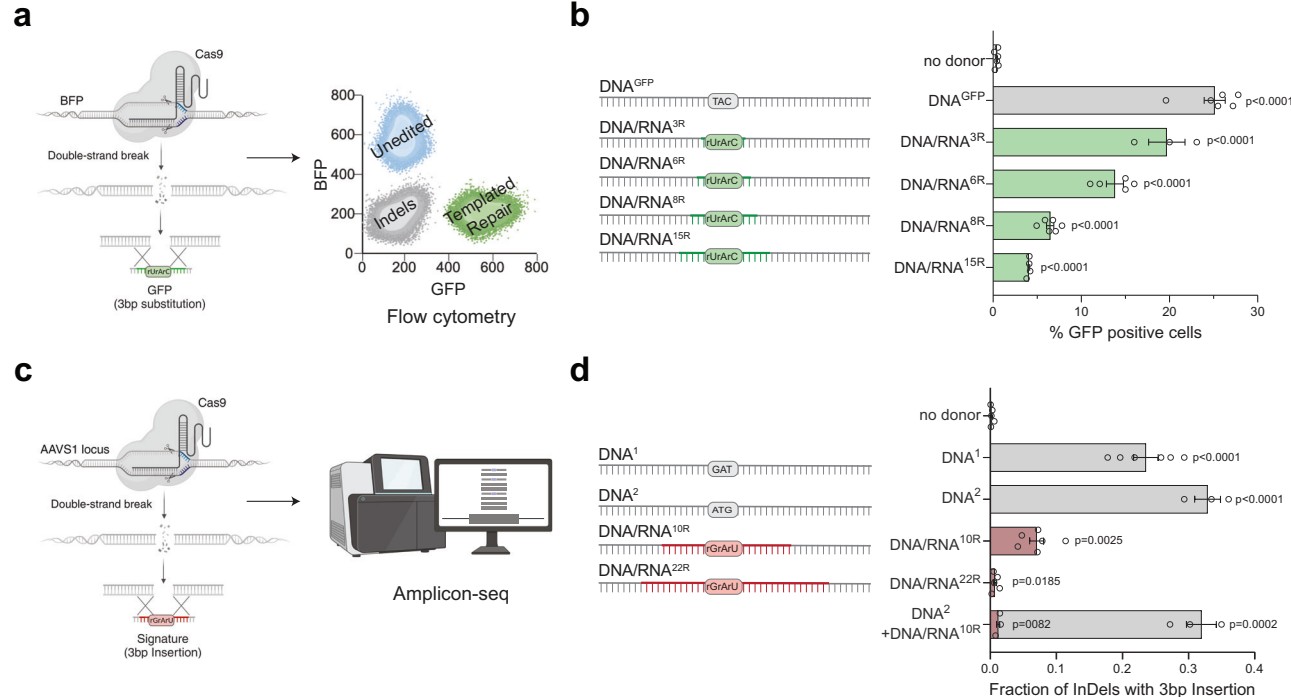

**Fig. 1 | Human cells use RNA to template DSB repair. a** Schematic of the BFP-to-GFP assay designed to generate a green fluorescent signal via RNA templated DSB repair (RT-DSBR). This assay exploits the single amino acid change that differentiates Blue Fluorescent Protein (BFP) from Green Fluorescent Protein (GFP), switching the fluorescence from blue to green. A DSB is introduced at an integrated BFP locus using CRISPR/Cas9, and cells repair the break with a single-stranded DNA donor (DNA$^{GFP}$) containing the GFP codon, switching from BFP to GFP fluorescence. To detect RT-DSBR activity, we used DNA/RNA chimeric donors in which the sequence required to swap the codon was encoded by ribonucleotides instead of deoxyribonucleotides. **b** Right: schematic of the 120 bp chimeric donors used in the BFP-to-GFP assay with green segments representing stretches of ribonucleotides. Left: GFP signal quantification was performed by flow cytometry with different donors (n = 3–6 biological replicates) and compared to a non-donor control. **c** Schematic of the AAVS1-seq assay. A targeted DSB is introduced at the *AAVS1*

genomic locus using CRISPR/Cas9, and the donor DNA or DNA/RNA chimeras containing a 3 bp insertion are transfected into the cells. Successful repair using the donor leads to the incorporation of the mutational signature, which is detected by PCR amplification and Next Generation Sequencing. **d** Right: a schematic of the 60 bp donor templates used in the AAVS1-seq assay, with red segments representing stretches of ribonucleotides. Left: quantification of the fraction of repair products containing the 3 bp insertion signature after the Cas9 DSB is repaired by different donors, as measured by the AAVS1-seq assay (n = 3–6 biological replicates) and compared to a non-donor control. For (**b** and **d**), Statistical significance was assessed using unpaired two-tailed *t*-tests. Error bars represent the standard error of the mean (± SEM). Schematics in Fig. 1a–c, and d were *Created in BioRender. (2025)* https://BioRender.com/9tmc1xb. Source data are provided as a Source Data file. See also Supplementary Fig. 1.

intermediate. Finally, we examined the repair efficiency of the DNA/RNA$^{10R}$ donor in the presence of a pure DNA donor (DNA$^2$) containing an alternative insertion (ATG). The competition experiment revealed substantial repair (GAT insertion), though at a reduced efficiency compared to the DNA/RNA$^{10R}$ donor alone (Fig. 1d), suggesting competition between the RT-DSBR and SSTR pathways. The two independent assays revealed that human cells possess reverse-transcriptase activity that copies RNA sequences embedded within a single-stranded oligonucleotide to mediate DSB repair.

## RT-DSBR operates independently of LINE-1 retrotransposon and Polθ

To determine whether known human reverse transcriptases participated in RT-DSBR, we investigated the role of LINE-1 retrotransposon and DNA polymerase theta (Polθ). LINE-1 retrotransposons are active in human cells, including HEK293T cells[23,34], and their mRNA has been detected at sites of DNA damage[35,36]. However, inhibiting LINE-1 reverse transcriptase activity using the HIV reverse transcriptase inhibitors azidothymidine (AZT) or lamivudine (3TC)[37] did not impair DSB repair as measured by BFP-to-GFP assay and AAVS1-seq, indicating that LINE-1 reverse transcriptase is dispensable for RT-DSBR (Fig. 2a, b and Supplementary Fig. 2a)[38]. We confirmed the efficacy of AZT and 3TC treatment using a fluorescent reporter for LINE-1 reverse transcription activity (Supplementary Fig. 2b)[39], which showed a three-fold reduction in integration following drug treatment (Supplementary Fig. 2c).

A recent study suggested that Polθ, which is critical for MMEJ, has reverse transcriptase activity in vitro, and that it can copy a donor template containing two rNTPs in vivo[25]. To assess the potential role of Polθ in RT-DSBR, we targeted *POLQ* using CRISPR/Cas9 to generate independent clonally derived *POLQ*$^{-/-}$ cells (Supplementary Fig. 2d, e). The BFP-to-GFP assay using DNA$^{GFP}$ and DNA/RNA$^{6R}$ donors revealed that *POLQ*$^{-/-}$ cells displayed a similar distribution of repair products compared to *POLQ*$^{+/+}$ cells and those rescued with full-length Polθ-FLAG (Fig. 2c and Supplementary Fig. 2d–f). In an independent set of experiments, we depleted Polθ using siRNA (Fig. 2d and Supplementary Fig. 2g, h). As expected, Polθ depletion reduced the MMEJ signature following DSB induction[31] (Supplementary Fig. 2i). However, Polθ loss did not impact RT-DSBR, as measured by the BFP-to-GFP and the AAVS1-seq assays (Fig. 2d and Supplementary Fig. 2d–f). Similarly, RT-DSBR was intact in cells treated with the small molecule inhibitor of Polθ, RP6685[40] (Supplementary Fig. 2j). Based on these findings, we concluded that LINE-1 reverse transcriptase and Polθ activity are dispensable for RT-DSBR.

## A targeted CRISPR/Cas9 screen highlights potential regulators of RT-DSBR

To identify factors that regulate RT-DSBR and to uncover the enzyme responsible for reverse transcription of the RNA moiety, we performed a targeted CRISPR/Cas9 screen using the BFP-to-GFP assay (Fig. 3a). We infected BFP-expressing cells with a focused library of sgRNAs

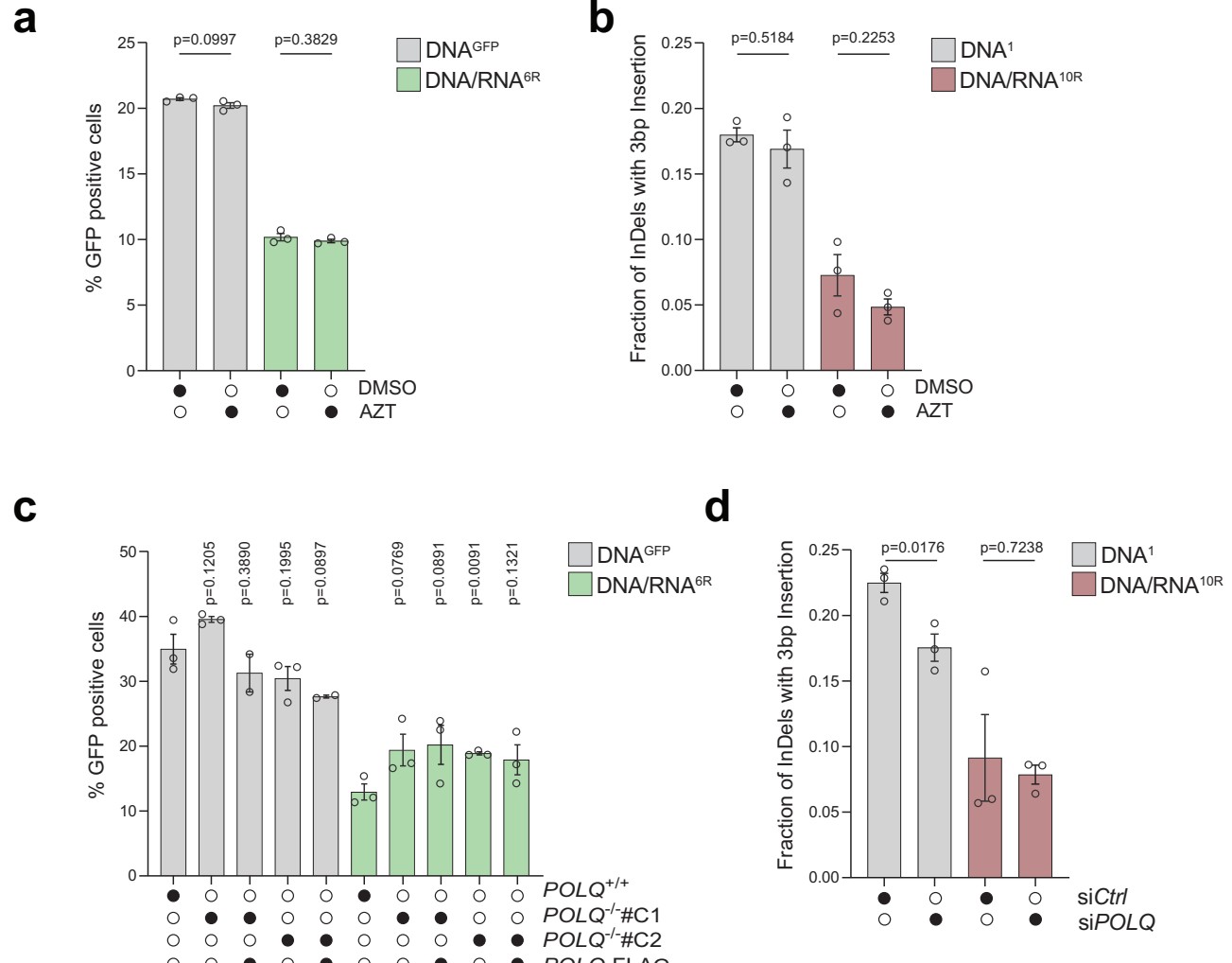

**Fig. 2 | RT-DSBR is independent of LINE-1 and Polθ activity. a** BFP-to-GFP assay with DNA$^{GFP}$ and DNA/RNA$^{6R}$ donors in the presence of 10 μM of the HIV reverse transcriptase inhibitor azidothymidine (AZT) or DMSO as a control ($n$ = 3 biological replicates). **b** AAVS1-seq performed with DNA$^1$ or DNA/RNA$^{10R}$ donors in the presence of 10 μM AZT or DMSO ($n$ = 3 biological replicates). **c** BFP-to-GFP assay with DNA$^{GFP}$ and DNA/RNA$^{6R}$ donors in two $POLQ^{-/-}$ clones (#C1 and #C2) with or without complementation by full-length $POLQ$ ($POLQ$-FLAG) ($n$ = 2 and 3 biological replicates). **d** AAVS1-seq with DNA$^1$ or DNA/RNA$^{10R}$ donors following knockdown of $POLQ$ through siRNA, compared to a non-targeting siRNA control (siCTRL) ($n$ = 3 biological replicates). For (**a–d**): Statistical significance was assessed using unpaired two-tailed $t$-tests. Error bars represent the standard error of the mean (± SEM). Source data are provided as a Source Data file. See also Supplementary Fig. 2.

targeting 1285 DNA damage response (DDR) genes. After ten days, we introduced a Cas9-gRNA RNP complex targeting the BFP locus with either DNA$^{GFP}$ or DNA/RNA$^{6R}$ donors. On day 14, we used FACS to isolate the templated-repair GFP$^+$ cells from the non-templated-repair GFP$^-$BFP$^-$ cells. We used NGS to compare gRNA abundance in GFP$^+$ and GFP$^-$BFP$^-$ populations and applied MAGeCK[41] to identify genes that inhibit RT-DSBR (enriched in GFP$^+$) or promote RT-DSBR (depleted in GFP$^+$). By comparing the initial ($t$ = 0 days) and final ($t$ = 14 days) time points, we confirmed stable gene knockdown and a robust hit calling based on the behavior of known essential genes (Supplementary Fig. 3a, b and Supplementary Data 2).

We identified several hits that promoted repair using DNA$^{GFP}$ or DNA/RNA$^{6R}$, including factors in the Fanconi anemia pathway (Fig. 3b). We also found that depletion of core factors involved in DNA end-resection (*NBN*, *EXO1*, *HELB*, *BRCA1*) and HR (*HELQ*, *BRCA1*, *RAD51B*, *RAD51AP1*) led to reduced DSB repair with both DNA and DNA/RNA donors (Fig. 3b, c and Supplementary Fig. 3c, d). These findings align with results from a similar CRISPR screen that used a DNA donor[30], suggesting that both Fanconi anemia and end-resection operate upstream of oligonucleotide-templated repair. On the contrary, the loss

of anti-resection factors, *TP53BP1*, *SHLD1*, and *KLHL15*, increased templated repair (Fig. 3b, c and Supplementary Fig. 3c, d). The latter observation is consistent with previous studies showing that *TP53BP1* depletion enhances CRISPR/Cas9 genome editing efficiency[42]. Interestingly, clonally derived *TP53BP1*$^{-/-}$ cells showed a three-fold increase in RT-DSBR when using the DNA/RNA$^{6R}$ donors compared to a *TP53BP1*$^{+/+}$ cell line and *TP53BP1*$^{-/-}$ cells complemented with 53BP1-FLAG (Supplementary Fig. 3e, f). In addition to hits that were common to both donor types, we identified genes that uniquely affected DSB repair using the ribonucleotide-containing donor (Fig. 3d). Top hits, including *hnRNPK* and *hnRNPC*, were validated as RT-DSBR factors using the BFP-to-GFP assay (Supplementary Fig. 3g, h). Given the role of RNA-binding proteins in mRNA maturation and splicing, they may facilitate the retention of the DNA/RNA donor at the site of the break. Alternatively, they may regulate the expression of genes required for RT-DSBR[43].

## The translesion synthesis polymerase zeta (Polζ) acts as a reverse transcriptase in vivo
To identify the reverse transcriptase responsible for RT-DSBR in human cells, we analyzed DNA polymerases based on their ranking in the

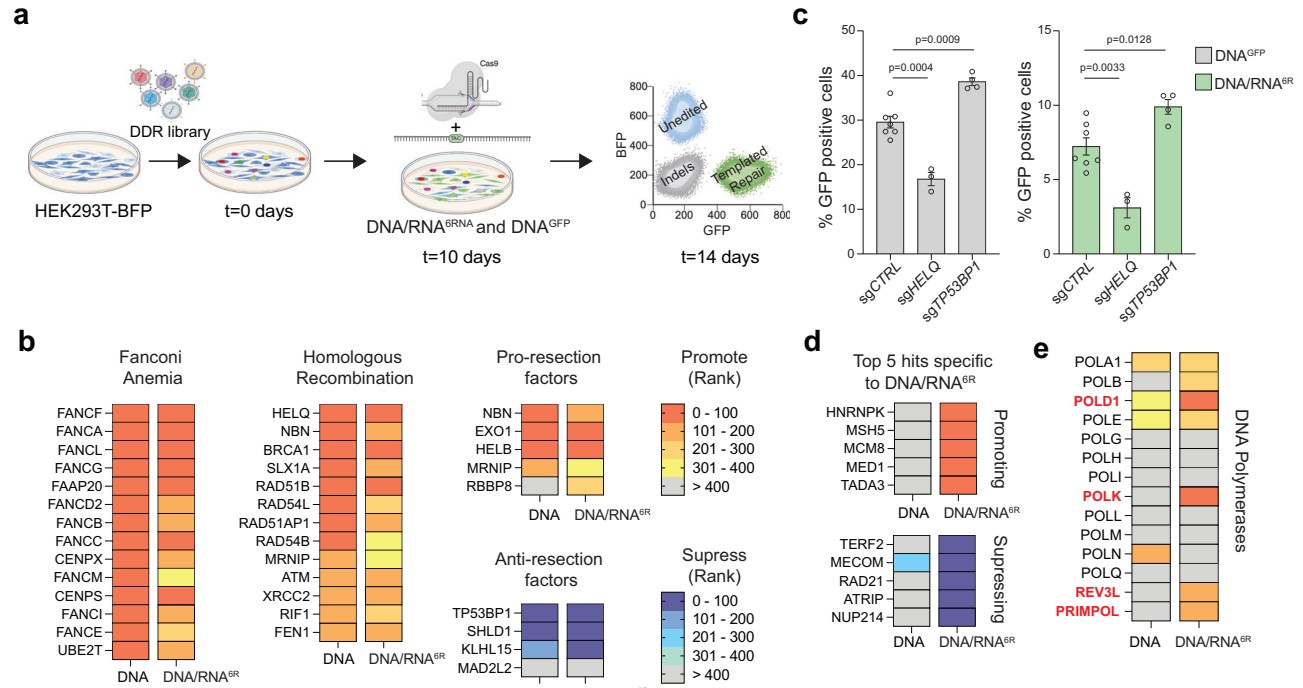

**Fig. 3 | A CRISPR/Cas9 screen identifies factors involved in RT-DSBR.**
**a** Schematic representation of a flow-based CRISPR/Cas9 screen performed using the BFP-to-GFP reporter in HEK293T cells. Cells were transduced with Cas9 and sgRNAs from a DNA damage library. After 10 days of sgRNA selection, the BFP-to-GFP assay was carried out using the DNA$^{GFP}$ or DNA/RNA$^{6R}$ donor respectively. **b** The CRISPR/Cas9 screen data were analyzed using the MAGeCK algorithm by comparing the GFP$^+$ sorted cells with the GFP$^-$ BFP$^-$ cells. A heatmap highlights selected genes with high-ranking scores, indicating factors that promote or suppress single-strand template repair. **c** BFP-to-GFP assay results

using DNA$^{GFP}$ and DNA/RNA$^{6R}$ donors after knockdown of two top hits that promote (*HELQ*) or suppress (*TP53BP1*) RT-DSBR ($n = 3–7$ biological replicates). sgRNA targeting the *AAVS1* locus was used as a control. Statistical significance was assessed using unpaired two-tailed $t$-tests. Error bars represent the standard error of the mean ($\pm$ SEM). **d** Heatmap of the 5 top hits that promote or suppress RT-DSBR. **e** Comparison of the rank position of major DNA polymerases identified in DNA$^{GFP}$ *vs.* DNA/RNA$^{6R}$ CRISPR/Cas9 screens. Schematic in Fig. 3a was *Created in BioRender. (2025)* https://BioRender.com/9tmc1xb. Source data are provided as a Source Data file. See also Supplementary Fig. 3.

CRISPR/Cas9 screen. Specifically, we compared the repair efficiency using the DNA/RNA versus DNA-only donors. Among the 14 human DNA polymerases evaluated, four ranked among the top 200 genes identified in the DNA/RNA donor screen and were less prominent in the DNA-only donor screen (Fig. 3e). These include the catalytic subunit of Polδ (*POLD1*) and Polζ (*REV3L*), *POLK*, and the primase *PRIMPOL*.

To explore the potential reverse transcriptase activity of these polymerases, we performed siRNA-mediated knockdowns and assessed RT-DSBR using the AAVS1-seq assay (Fig. 4a). We targeted three additional polymerases: Polη (*POLH*), previously shown to have reverse transcriptase activity in vitro and in vivo[44–46], Polμ (*POLM*) which incorporates ribonucleotides at break sites before ligation[47,48], and Polν (*POLN*) a member of the same family as Polθ[45,49]. Among the seven polymerases tested, knockdowns of *REV3L* and *POLD1* showed a reduction in RT-DSBR (Fig. 4a and Supplementary Fig. 4a). However, *POLD1* knockdown also reduced repair events mediated by the DNA donor, indicating that POLD1 is not specific to RT-DSBR. In contrast, *REV3L* knockdown did not affect repair through the DNA donor, suggesting its specificity in reverse transcribing the RNA template during DSB repair. Consistently, the depletion of *REV3L* led to a significant reduction in RT-DSBR measured by the BFP-to-GFP assay (Fig. 4b and Supplementary Fig. 4b, c). Cell cycle analysis showed no change in the distribution of cells in S-phase following the depletion of *REV3L*, ruling out a cell cycle effect of the knock-down (Supplementary Fig. 4d).

Polζ is a multi-subunit complex comprising the catalytic core REV3L and the accessory subunits POLD2, POLD3, and REV7, which interact with the DNA transferase REV1[50] (Fig. 4c). Consistent with the role of the Polζ complex in reverse transcribing RNA to DNA, depletion

of *POLD3*, *REV1*, and *REV7* subunits also led to a reduction in RT-DSBR at the *AAVS1* locus using the DNA/RNA donor (Fig. 4d and Supplementary Fig. 4e). This finding aligns with the CRISPR/Cas9 screen, in which the Polζ complex subunits rank higher when using the chimera donor compared to the DNA donor (Supplementary Fig. 4f). Taken together, our results suggest that Polζ is a key reverse transcriptase involved in RT-DSBR.

## Transcript RNA serves as a template for polymerase ζ-dependent RT-DSBR

Our reporter assays revealed that human cells can utilize synthetic oligonucleotides containing RNA as templates for DSB repair (Fig. 1). This prompted us to investigate whether an RNA transcript could serve as a template for RT-DSBR mediated by Polζ. To that end, we amended the AAVS1-seq assay by introducing an mRNA transcribed from a plasmid as the donor template. This mRNA encodes the *AAVS1* sequence containing a three-base pair (GAT) insertion interrupted by the human beta-globin intron[51] (Fig. 4e, f). The insertion, spanning the splice junctions, allowed us to differentiate between repair events via the spliced RNA transcript and those mediated by copying the donor plasmid. We verified the correct transcript splicing through PCR analysis using primers spanning the splice sites (Supplementary Fig. 4g, h). NGS analysis of the amplicon sequence revealed a small fraction of repair events containing the GAT insertion sequence (Fig. 4g). Significantly, no additional insertion signatures were associated with the presence of the transcript RNA, confirming that the GAT insertion was specific to RT-DSBR activity (Supplementary Fig. 5). When *REV3L* was depleted, the characteristic GAT insertion signature associated with

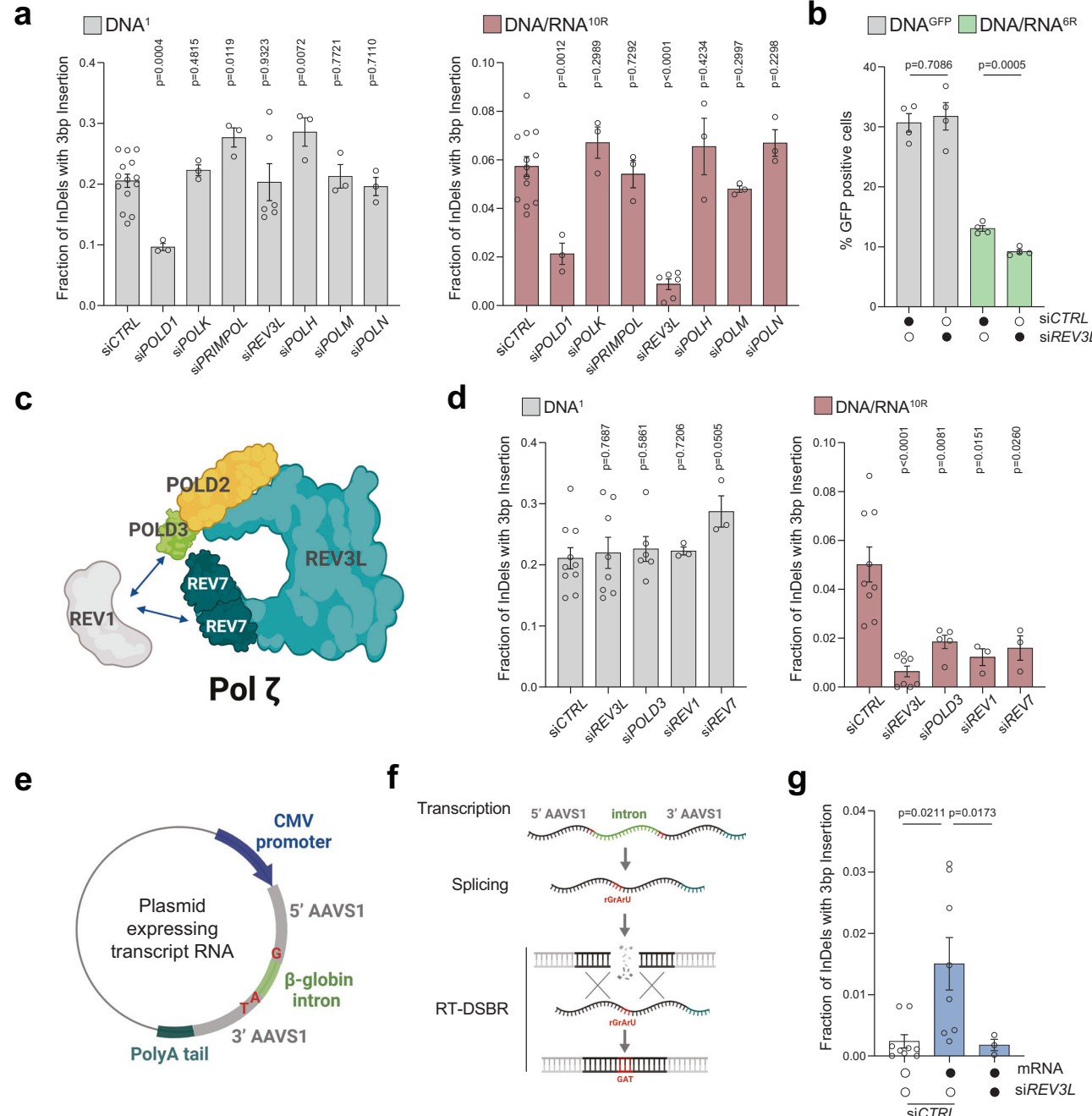

**Fig. 4 | Transcript RNA is a donor for DNA polymerase zeta (ζ) dependent RT-DSBR. a** Fraction of repair products from AAVS1-seq using DNA[1] or DNA/RNA[10R] donors after siRNA-mediated knockdown of *POLD1, POLK, PRIMPOL, REV3L, POLH, POLM* or *POLN*, compared to a non-targeting siRNA control (siCTRL) ($n = 3$–14 biological replicates). **b** Percentage of repair products from the BFP-to-GFP assay using DNA[GFP] or DNA/RNA[6R] donors following knockdown of *REV3L* with siRNA. **c** Schematic of the Polζ complex. **d** Effect of Polζ subunits depletion on AAVS1-seq repair outcomes with DNA[1] or DNA/RNA[10R] donors, assessed after siRNA-mediated knockdown ($n = 3$–9 biological replicates). **e**, **f** Schematic of a plasmid-based system designated to generate transcript RNA that acts as a donor template. Homology arms (grey) flank the Cas9 break site at the *AAVS1* locus. Blue: CMV promoter.

Light green- β-globin: artificial intron. Dark green: poly-A tail. Red: insertion signature. **g** Fraction of repair products containing the mutational signature in the presence of no donor ($n = 9$ biological replicates) or transcript RNA donor, following Cas9-induced breaks. Data were collected after treatment with non-targeting siRNA (siCTRL) ($n = 8$ biological replicates) or siRNA against *REV3L* ($n = 3$ biological replicates). For (**a**–**g**): Where applicable, statistical significance was assessed using unpaired two-tailed *t*-tests, with Welch's correction in (**g**). Error bars represent the standard error of the mean (± SEM). Schematics in this figure (**c**, **e**, and **f**) were *created in BioRender. (2025)* https://BioRender.com/9tmc1xb. Source data are provided as a Source Data file. See also Supplementary Figs. 4 and 5.

RT-DSBR was significantly reduced (Fig. 4g). In conclusion, our data suggest that human cells can use a spliced mRNA complementary to the damage site as a template for DSB repair. Furthermore, this process depends on the Polζ complex, underscoring its essential role in RNA-templated DSB repair.

## Whole intron deletion, a genomic scar reflective of RT-DSBR in human cancers

So far, our experiments have shown that RNA can serve as a template for repairing CRISPR/Cas9-induced DSBs in human cells. These findings suggest that mRNA transcripts at naturally occurring endogenous

break sites might provide a template for DSB repair. However, detecting RT-DSBR at endogenous breaks poses a challenge because RNA-mediated repair typically leaves no detectable scar. An exception would occur if a spliced mRNA transcript were used to repair a break within an intron. In such cases, RT-DSBR could create a distinct signature by precisely removing the intron from the genome, resulting in a whole intron deletion (WID) event (Fig. 5a). Although WIDs are expected to be rare, they are potentially reflective of RT-DSBR activity.

To investigate whether RT-DSBR can lead to a WID in cells, we targeted a small intron of a highly transcribed gene (*CALR*)[52] using a CRISPR/Cas9-induced break in clonally derived *TP53BP1*[-/-] cells. Analysis by TIDE confirmed efficient cleavage within the intron (Supplementary Fig. 6a). As a positive control, alongside the CRISPR/Cas9 targeting the *CALR* intron 2, we co-transfected cells with an oligonucleotide designed to mimic a WID event using a donor template. The donor oligonucleotide comprised a DNA/RNA chimera lacking the intronic sequence but complementary to the adjacent exon sequences and containing six ribonucleotides spanning the exon-exon junction. We amplified repair products using primers specific to the flanking exons. Subsequent NGS analysis using CRISPResso2 identified a subset of repaired sequences exhibiting the precise deletion of the second intron as mimicked in the donor (Supplementary Fig 6b– left graph). To test the hypothesis that endogenous spliced *CALR* mRNA could serve as a repair template, we transfected the sgRNA targeting *CALR* intron 2 without providing the exogenous donor template. We detected a low but statistically significant accumulation of WID events (Fig. 5b and Supplementary Fig. 6b–right graph). Consistent with Polζ promoting reverse transcription using mRNA, WID events at the *CALR* locus were significantly reduced upon depletion of *REV3L* using siRNA (Fig. 5b and Supplementary Fig. 6c, d). Inhibiting LINE-1 reverse transcriptase activity via AZT did not impair WID events through this assay, again indicating that LINE-1 reverse transcriptase is dispensable for RT-DSBR (Supplementary Fig. 6e, f). We observed similar WID events driven by endogenous transcripts when targeting another highly transcribed gene—*GNAS*—with a CRISPR/Cas9-induced break at intron 11 (Supplementary Fig. 6g-h), but not the transcriptionally silent gene, *IL3*, at intron 4 (Supplementary Fig. 6i, j). These findings suggest RT-DSBR can lead to WIDs when using spliced mRNA as the template to repair a break.

Next, we examined the repertoire of genomic alterations in cancer genomes from tumor samples, available through MSK-IMPACT and PCAWG[26–28], to determine whether spliced mRNA could produce intron deletions in tumor cells from naturally occurring endogenous DNA damage. MSK-IMPACT is a hybridization capture-based sequencing assay that analyzes matched tumor/normal samples, covering all coding and selected intronic or regulatory regions of at least 341 essential cancer genes[26]. To identify WIDs, we systematically screened for somatic deletions in 64,544 tumors (from 56,322 patients) that underwent MSK-IMPACT sequencing. By aligning these deletions to the reference genome, we identified 113 unique deletions precisely spanning intronic sequences classified as WIDs (Fig. 5c and Supplementary Data 3.1). We examined RNA-seq data from the identified tumors as a control, confirming that genes with WIDs are actively transcribed (Supplementary Fig. 6k and Supplementary Data 3.2). We validated the presence of WIDs in two independent genes (*HLA-B* and *GNAS*) in patient-derived tumor samples through PCR amplification of a region spanning the deleted introns, followed by Sanger sequencing (Fig. 5d–g, Supplementary Fig. 6l–n, and Supplementary Data 1.7). To further corroborate our findings from MSK-IMPACT, we conducted an independent analysis using WGS data from PCAWG[28], which contains data from 1902 patients and tumor samples, with matched normal tissues across 38 tumor types (Supplementary Fig. 6o)[28]. This analysis revealed 16 additional WIDs, supporting the detection of RT-DSBR activity in a second well-known cancer genome database (Supplementary Fig. 6o, p and Supplementary Data 3.3).

Given the paucity of WID events, it was essential to rule out that their occurrence was due to chance. We conducted a simulation analysis involving 10,000 cohorts of the study genomes, estimating the number of WIDs expected from random deletion events. Each cohort contains a similar number of deletions observed in MSK-IMPACT, with deletions randomly distributed across the genomes while considering deletion lengths and gene content. Although the overall distribution of random deletions closely resembled that pattern seen in MSK-IMPACT, the maximum number of WID events observed across 10,000 simulated cohorts was only four, which occurred in just two cohorts (Fig. 5h–j). These findings suggest that the likelihood of observing 113 WIDs in the MSK-IMPACT data by chance is negligible ($p < 0.0001$) (Fig. 5j). We conducted a similar simulation analysis on data from the PCAWG project, which further confirmed that the observed WID events in PCAWG are also unlikely to have occurred by chance ($p < 0.0001$) (Supplementary Fig. 6q, r).

Furthermore, having observed that of the total 113 WIDs identified in MSK-IMPACT, we found approximately half of the deletions occur in clusters of two or more consecutive WIDs, with some genes losing as many as five consecutive introns (Fig. 6a, b, Table 1, and Supplementary Data 3.1). Canonical DSB repair is highly unlikely to lead to the loss of one, let alone sequential introns, as this would require multiple breaks in adjacent introns to occur. The presence of consecutive WIDs provides further evidence that these introns are lost due to using a spliced mRNA as a donor template, which would lack consecutive introns when used. Moreover, our analysis likely underestimates the number of detected genes with >2 consecutive WIDs due to the limitations of the deletion callers in detecting large deletions owing to the sequencing methods used in MSK-IMPACT[27]. This is observed at genes like *XPO1* and *JAK1*, where we detected two or more consecutive intron losses separated by a single remaining large intron (Table 1). Finally, we confirmed these observations of consecutive WIDs by analyzing adjacent WID events following the CRISPR/Cas9-mediated cleavage of intron 2 in *CALR*. Notably, there was a significant accumulation of WID events upstream of the break sites (Fig. 6c). Our findings implicate RT-DSBR activity in repairing breaks at actively transcribed genes through endogenous spliced mRNA and provide a plausible mechanism for this genomic scar (Fig. 6d).

## Discussion

Emerging evidence suggests that RNA transcripts can indirectly shape the landscape of DSB repair by modulating three canonical repair pathways: HR, NHEJ, and MMEJ[17,53]. Transcription and DNA repair are intrinsically linked processes, as evident by the evolution of transcription-coupled nucleotide excision repair, a specialized DNA repair pathway[54]. Moreover, the annealing of RNA with the complementary strand of DNA to form an R-loop can act as a scaffold that recruits repair factors and increases HR efficiency. This effect is pronounced in highly transcribed genes, thus providing evidence of a role for RNA in modulating the outcome of DSB repair[16,54]. Despite this, understanding whether RNA can directly serve as a template for DSB repair has been challenging due to the lack of tools to assess its contribution in higher eukaryotes. In this study, we demonstrate that RNA serves as a template for DSB repair via reverse transcription facilitated by the DNA polymerase ζ complex. We show that RT-DSBR using mRNA is a rare mutagenic pathway in human tumors with a highly characteristic WID genomic scar. Given the abundance of RNA and its encoding of genetic information, utilizing RNA to restore lost genetic information following DSBs may be potentially driven by selective pressure to preserve the integrity of highly transcribed genes.

### Transcript RNA as a template for DSB repair in human cells

Based on our findings, we propose a model in which DSBs occurring in actively transcribed genes can utilize the corresponding RNA transcript as a template for repair (Fig. 6d). It remains unclear whether the RNA

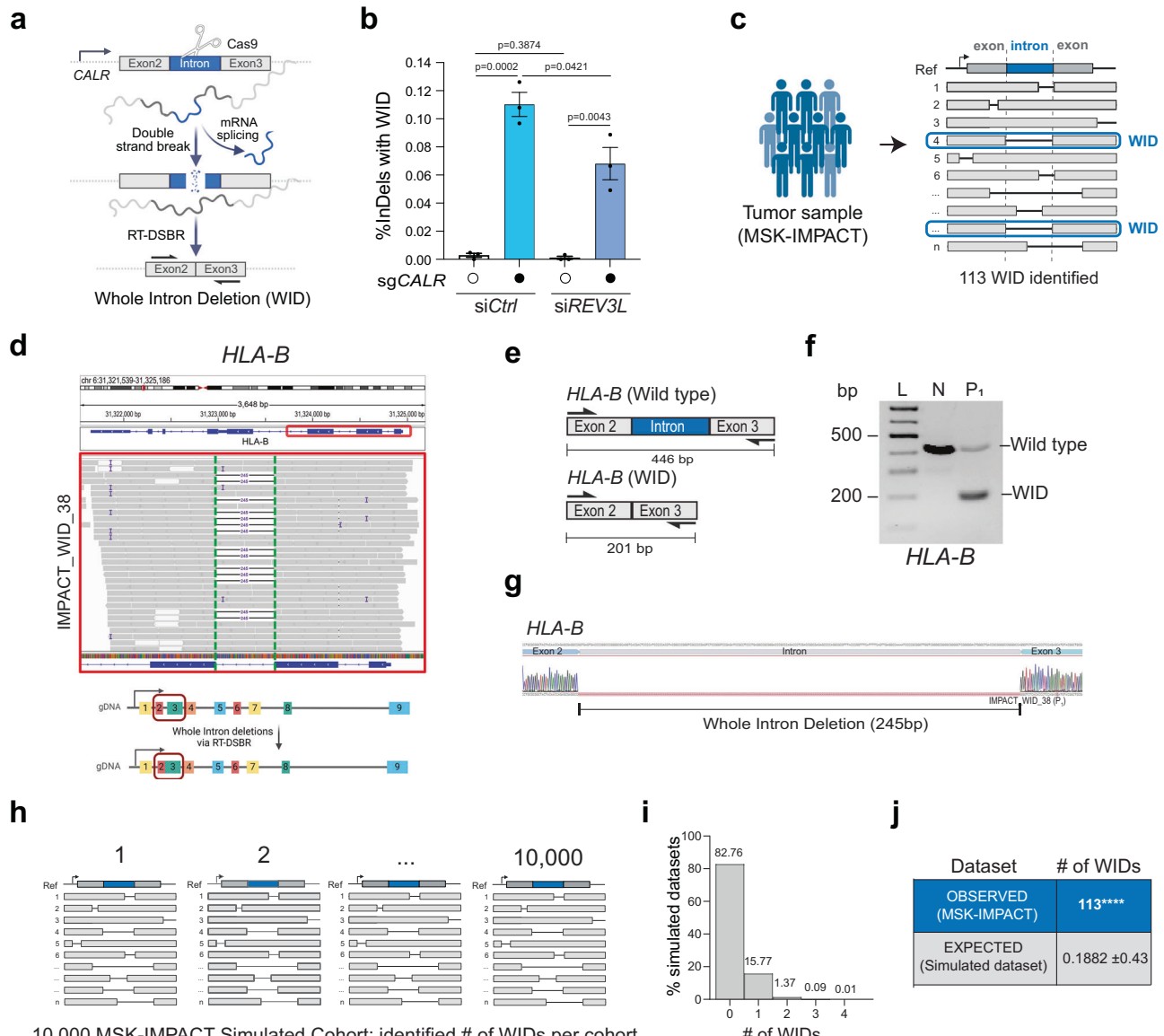

**Fig. 5 | Whole intron deletions from cancer genomes provide in vivo evidence of RT-DSBR. a** Schematic of the CRISPR/Cas9 assay to detect a whole intron deletion (WID) in human cells. **b** Quantification of reads containing precise WIDs (as a fraction of total repair events) at *CALR* intron 2 in control cells and ones treated with siREV3L (*n* = 3 biological replicates) with and without a CRISPR/Cas9-mediated DSB. Statistical significance was assessed using an unpaired two-tailed *t*-tests. Error bars represent the standard error of the mean (± SEM). **c** Schematic of the bioinformatic pipeline used to analyze deletions in tumors from the MSK-IMPACT database. WIDs were identified as deletions that span a precise entire intron. The blue box highlights a read showing perfect intron loss. **d** Example of a WID found in the *HLA-B* gene of a patient sample from the MSK-IMPACT cohort. Read bases that match the reference are displayed in gray, purple "I" represents insertions, and deletions are indicated with a black dash (−). Alignments displayed with light gray borders and white fill have a mapping quality equal to zero, suggesting they may map to multiple regions across the genome. A 245 bp deletion is observed upon targeted NGS that maps precisely to the area corresponding to the intron flanked

by Exon 2–3 of the *HLA-B* gene. **e** Schematic of the exons spanning the WID in *HLA-B* with the flanking primers used to confirm the sequence. **f** Agarose gel depicting the full-length band corresponding to the locus spanning Exon 2–3 in normal MCF-12A cells (N) and the shorted locus with the intron loss in the tumor sample in *HLA-B*. P₁ represents a patient from the MSK-IMPACT cohort (*n* = 1 biological replicate). **g** Sanger sequencing of the PCR products to confirm the presence of the WID in *HLA-B*. **h** Graph representing the number of WID observed in the simulated datasets (10,000 MSK-IMPACT-like cohorts). **i, j** Total number of WIDs over 73,030 total deletions identified in 64,544 tumor samples of the MSK-IMPACT database. The number of expected WIDs was calculated after randomization of the deletion locations across the whole genome. Using two-tailed Fisher's exact test, empirical *p*-values were calculated by comparing the observed versus the 10,000 random values (**** *p* < 0.0001). Schematics in this figure (**a, c–e,** and **h**) were *created in BioRender. (2025)* https://BioRender.com/9tmc1xb. Source data are provided as a Source Data file. See also Supplementary Fig. 6.

transcript used for repair is generated before or after DSB formation. However, since transcription is disrupted in response to DNA damage[4–6], we favor a scenario in which the donor RNA template is transcribed before DSB formation. Once the RNA anneals to the processed DNA end, we demonstrate that PolζZ can use the mRNA to fill the gap via reverse transcription, restoring the original genetic information.

Recently, other translesion polymerases, specifically Polη and Polθ, have exhibited reverse transcriptase activity both in vitro and in vivo[25,46]. However, our assays could not detect reverse transcription activity for Polη and Polθ (Figs. 2 and 4). In addition to Polζ, our CRISPR/Cas9 screen identified 53BP1 as a factor that suppresses RT-DSBR, which we validated in a *TP53BP1⁻/⁻* clone. Since 53BP1 is known to

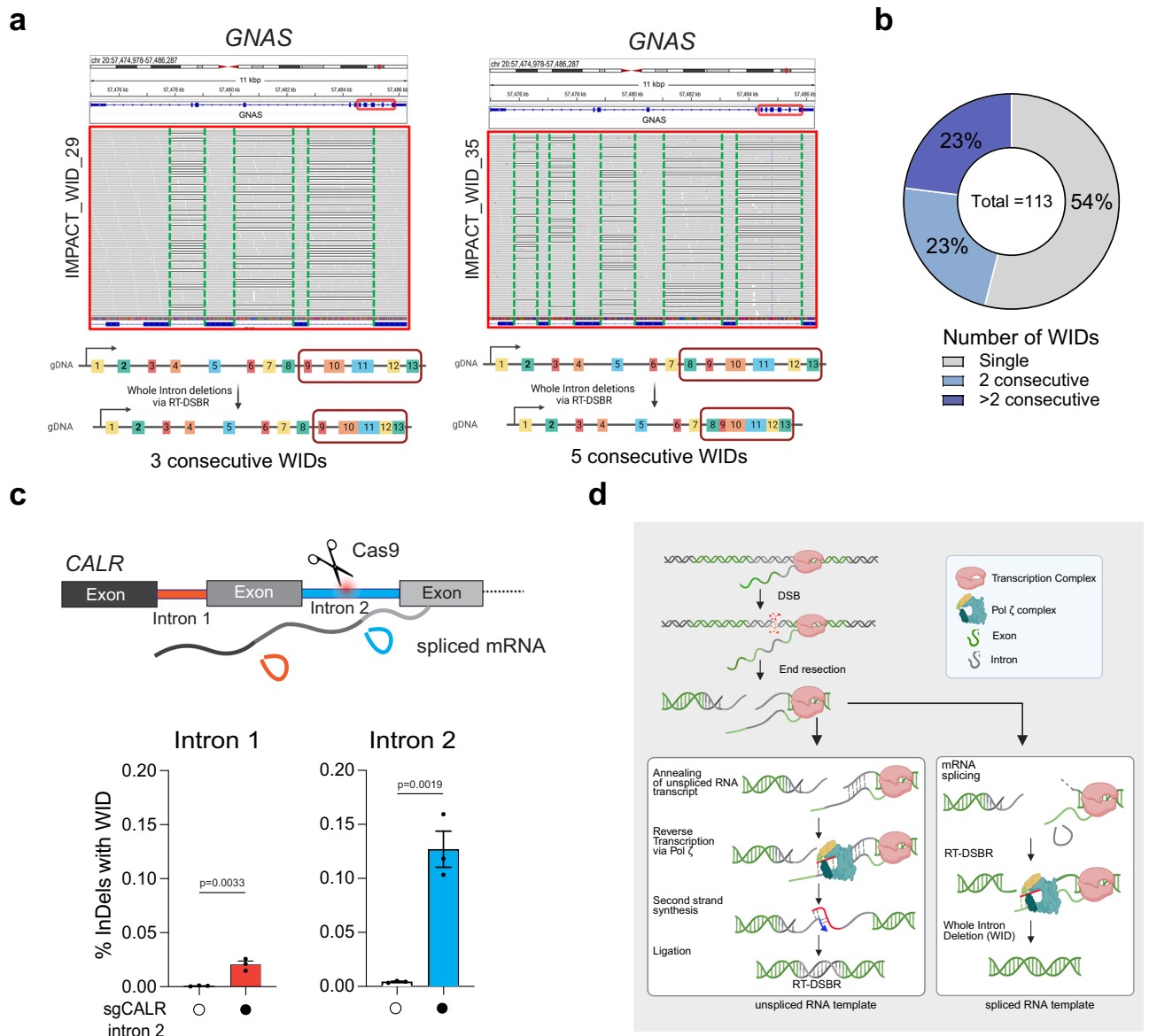

**Fig. 6 | Further evidence of Whole Intron deletions in cells and tumors.**
**a** Example of consecutive WIDs detected in the *GNAS* gene from patient samples sequenced with MSK-IMPACT. Grey bases match the reference genome, and deletions are indicated with a black dash (−). On the left, three deletions were observed at the *GNAS* gene following targeted NGS, precisely mapping the introns flanked by Exon 10–11, 11–12, and 12–13, respectively. On the right, five consecutive deletions were mapped to introns flanked by Exon 8–9, 9–10, 10–11, 11–12, and 12–13.
**b** Frequency of consecutive WIDs observed in the MSK-IMPACT dataset. **c** Loss of upstream intron following cleavage of *CALR* intron 2 with CRISPR/Cas9. Top, schematic representation of multiple introns in *CALR* gene with cleavage of intron 2. The bottom graph depicts the quantification of reads containing WIDs in an intron adjacent to the cleavage site (*n* = 3 biological replicates). Statistical

significance was assessed using an unpaired two-tailed *t*-tests. Error bars represent the standard error of the mean (± SEM). **d** Proposed model for RT-DSBR: When a double-strand break (DSB) occurs within an actively transcribed gene, the existing RNA transcript base-pairs with the cleaved template strand and is reverse transcribed by the Polζ complex. The newly synthesized DNA (shown in red) anneals to the resected opposite end, facilitating second-strand synthesis, gap filling, and ligation. The specific polymerase and ligase involved in this process have yet to be identified. If a spliced RNA transcript serves as the repair template, the intronic sequence will be omitted, resulting in a genetic scar known as a whole intron deletion (WID). Schematic in this figure (**d**) *was created in BioRender. (2025)* https://BioRender.com/9tmc1xb. Source data are provided as a Source Data file.

counteract DNA resection[55], resection may be a crucial step required to process the breaks before the annealing of the RNA template. This genetic manipulation, which increases the use of RT-DSBR, could facilitate the identification of other factors involved in this pathway.

### Conservation of RNA-templated DSB repair from yeast to humans
RT-DSBR appears to be a conserved mechanism from yeast to humans[18,19]. In *S. cerevisiae*, Ty1 retrotransposons mediate cDNA synthesis from mRNA for DSB repair via an HR-like mechanism. In the

absence of Ty1, Polζ reverse transcribes the RNA at break sites in cis to mediate repair[20]. Unlike in yeast, LINE-1 retrotransposon is dispensable for RT-DSBR in human cells (Fig. 2 and Supplementary Fig. 6). Instead, Polζ has a prominent role in copying the RNA to mediate DSB repair. Furthermore, as opposed to RT-DSBR in yeast, which was detected only in the absence of RNaseH1 and RNaseH2, we detect low but significant RNA templated repair in human cells competent for both enzymes (Figs. 1 and 4d, g). While these results suggest that RNA:DNA hybrid removal is not essential for RT-DSBR in human cells, whether RNaseH1 and RNaseH2 have a role in this process remains unexplored.

**Table 1 | Evidence of consecutive WIDs in tumors**

| Patient ID from this study | Gene | Left exon | Right exon | deletion size observed (bps) | intron size (bps) | Total no of consecutive WIDs |
|---|---|---|---|---|---|---|
| IMPACT_WID_45 | DICER1 | exon27 | exon26 | 370 | 370 | 2 |
| | | exon26 | exon25 | 93 | 93 | |
| IMPACT_WID_42 | SF3B1 | exon20 | exon19 | 85 | 85 | 2 |
| | | exon19 | exon18 | 280 | 280 | |
| | SF3B1 | exon17 | exon16 | 216 | 216 | 2 |
| | | exon16 | exon15 | 96 | 96 | |
| IMPACT_WID_26 | XPO1 | exon23 | exon22 | 417 | 417 | 2 |
| | | exon22 | exon21 | 845 | 845 | |
| IMPACT_WID_36 | PIK3CA | exon18 | exon19 | 561 | 561 | 2 |
| | | exon19 | exon20 | 103 | 103 | |
| IMPACT_WID_32 | JAK1 | exon25 | exon24 | 738 | 738 | 2 |
| | | exon24 | exon23 | 591 | 591 | |
| | JAK1 | exon22 | exon21 | 360 | 360 | 2 |
| | | exon21 | exon20 | 1013 | 1013 | |
| IMPACT_WID_05 | ERRFI1 | exon4 | exon3 | 911 | 911 | 2 |
| | | exon3 | exon2 | 110 | 110 | |
| IMPACT_WID_11 | CSDE1 | exon13 | exon12 | 877 | 877 | 2 |
| | | exon12 | exon11 | 596 | 596 | |
| IMPACT_WID_21 | DAXX | exon7 | exon6 | 194 | 194 | 2 |
| | | exon6 | exon5 | 160 | 160 | |
| IMPACT_WID_20 | CSDE1 | exon13 | exon12 | 877 | 877 | 2 |
| | | exon12 | exon11 | 596 | 596 | |
| IMPACT_WID_12 | GNAS | exon11 | exon12 | 252 | 252 | 2 |
| | | exon12 | exon13 | 281 | 281 | |
| IMPACT_WID_04 | CDK8 | exon10 | exon11 | 718 | 718 | 2 |
| | | exon11 | exon12 | 118 | 118 | |
| IMPACT_WID_34 | GNAS | exon10 | exon11 | 146 | 146 | 3 |
| | | exon11 | exon12 | 252 | 252 | |
| | | exon12 | exon13 | 281 | 281 | |
| IMPACT_WID_29 | GNAS | exon10 | exon11 | 146 | 146 | 3 |
| | | exon11 | exon12 | 252 | 252 | |
| | | exon12 | exon13 | 281 | 281 | |
| IMPACT_WID_22 | GNAS | exon10 | exon11 | 146 | 146 | 3 |
| | | exon11 | exon12 | 252 | 252 | |
| | | exon12 | exon13 | 281 | 281 | |
| IMPACT_WID_09 | GNAS | exon10 | exon11 | 146 | 146 | 3 |
| | | exon11 | exon12 | 252 | 252 | |
| | | exon12 | exon13 | 281 | 281 | |
| IMPACT_WID_26 | XPO1 | exon16 | exon15 | 126 | 126 | 4 |
| | | exon15 | exon14 | 85 | 85 | |
| | | exon14 | exon13 | 166 | 166 | |
| | | exon13 | exon12 | 840 | 840 | |
| IMPACT_WID_19 | GNAS | exon8 | exon9 | 97 | 97 | 5 |
| | | exon9 | exon10 | 104 | 104 | |
| | | exon10 | exon11 | 146 | 146 | |
| | | exon11 | exon12 | 252 | 252 | |
| | | exon12 | exon13 | 281 | 281 | |
| IMPACT_WID_35 | GNAS | exon8 | exon9 | 97 | 97 | 5 |
| | | exon9 | exon10 | 104 | 104 | |
| | | exon10 | exon11 | 146 | 146 | |
| | | exon11 | exon12 | 252 | 252 | |
| | | exon12 | exon13 | 281 | 281 | |

A list of the patient samples exhibiting consecutive WIDs in the same gene, including the locations and sizes of the deletions.

Polζ is a critical translesion synthesis (TLS) polymerase responsible for synthesizing across various types of DNA lesions, including abasic sites and UV-damaged bases[50]. In contrast to other TLS polymerases, Polζ belongs to the B family of DNA polymerases, which includes accurate replicative polymerases. However, Polζ lacks 3′-5′ exonucleolytic proofreading activity, contributing to spontaneous mutagenesis in eukaryotic cells[56]. Notably, Polζ was reported to bypass single ribonucleotides in yeast, preventing replication fork stalling. In vitro studies have shown that the catalytic subunit of Polζ can efficiently bypass four ribonucleotides in tandem, highlighting its potential reverse transcriptase activity[57,58]. Deleting *REV3L* in chicken or mammalian cells causes hypersensitivity to genotoxic stress, including agents that induce DSBs[59,60]. Our findings highlight a previously unknown function of mammalian Polζ to copy RNA into DNA during RT-DSBR. The mechanisms by which Polζ is recruited to DSBs and regulated at these sites remain unknown. Additionally, future efforts exploring whether transcription influences its recruitment to DSBs may provide further insights into its role in RT-DSBR.

### Whole intron deletions: a genomic signature of RT-DSBR

Our model predicts that DSB repair can occur without leaving a detectable scar when pre-spliced RNA transcripts are used as templates. As such, detecting RT-DSBR activity in higher eukaryotes is particularly challenging because, in most cases, it leaves no genomic signature. However, when the RNA donor has already undergone splicing, repair of a break within an intron would lead to intron removal from the genome. In such cases, reverse transcription of the spliced RNA could result in genetic scars, such as WIDs, providing evidence of RT-DSBR activity in vivo. We provide evidence of intron loss by inducing a CRISPR/Cas9-break in introns of highly transcribed genes (Fig. 5a, b). Our data show the accumulation of WIDs in human tumor samples, indicating that DSBs can be repaired using spliced mRNA. The low frequency of WIDs in tumors limits our ability to determine whether specific mutations or genomic features influence this pathway and contribute to intron loss. The detection of a cluster of 2 or more consecutive and precise WIDs (Fig. 6) strongly indicates the use of RT-DSBR. This scenario can only be explained by spliced mRNA serving as a template for RT-DSBR, especially since other repair mechanisms are unlikely to result in the loss of sequential introns with precise exon-exon junctions.

Although WIDs are rare in tumors, we cannot exclude the possibility that intron loss events also occur in normal cells. Phylogenetic studies comparing genomes of organisms with abundant introns to those with fewer introns reveal a bias towards 3′ end intron loss. Two primary hypotheses have been suggested to explain this bias: one theory, based on studies in *Caenorhabditis elegans* and *Drosophila melanogaster*, posits that intron loss results from error-prone DSB repair by MMEJ and is driven by sequence homology near the ends of the break site[61]. An alternative hypothesis suggests that intron loss is due to retrotransposon-mediated reverse transcription of spliced mRNA[62–64]. Our data indicate that neither human retrotransposon activity nor MMEJ is involved in RT-DSBR-dependent intron loss. Instead, we show that Polζ-mediated RT-DSBR is active in human cells and can spontaneously produce intron loss following a DSB in an intron. Whether RT-DSBR contributed to intron loss during evolution remains to be determined. In a related context, it has been suggested that some pseudogenes form when mRNA transcripts are reverse-transcribed by LINE-1 and integrated into new locations in the genome[65]. These processed pseudogenes lack introns and may be driven by RT-DSBR activity in the germline.

### RNA-templated DSB repair under physiological conditions

With ~78% of the human genome actively transcribed[3], RT-DSBR may be more common in transcriptionally active loci, especially in contexts where homologous DNA templates are absent. Specifically, RT-DSBR may offer an error-free repair system for active genes in non-dividing cells, where error-free HR is blocked, and NHEJ is the only available option for DSB repair. For example, in neuronal cells, topoisomerase II-induced DSBs are stimulated by neuronal activity to resolve topological constraints at highly transcribed genes[66,67]. In such settings, RT-DSBR may offer a safer alternative to NHEJ for repairing these physiological breaks in non-dividing cells, thus safeguarding the genome. Selective pressures may have favored the development of such mechanisms to maintain genomic integrity, for example, at highly expressed loci, which are vital for cellular homeostasis.

In summary, our findings provide new insights into the role of transcript RNA as a template for DSB repair, highlighting a link between transcription and DSB repair in human cells. Further studies in diverse biological contexts will unravel the full spectrum of RT-DSBR activity and its implications for genome stability and evolution.

## Methods

### Cell culture

HEK293T (ATCC, #CLR-3216) cells were routinely grown with Dulbecco's Modified Eagle's Medium media supplemented with 10% bovine calf serum, 100 U/mL Penicillin-Streptomycin, 1% non-essential amino acids, 2mM L-glutamine. Cells were grown in a 37 °C and 5% CO₂ air incubator. Cells stably expressing the BFP reporter underwent selection with 20 μg/mL Blasticidin for ten days and then sorted with FACS. To generate *POLQ* KO clones and *TP53BP1* KO clones, $10^6$ HEK293T-BFP cells were transfected with Cas9 expressing plasmid and two gRNAs targeting exon 3 of *POLQ* or one gRNA targeting exon 6 of *TP53BP1*, respectively (Supplementary Data 1.5). Individual colonies were seeded into a 96-well plate and grown to confluence before proceeding with genotyping and western blotting. For azidothymidine (AZT) or lamivudine (3TC)-treated samples, cells were treated for 24 h with 10 μM of AZT (MilliporeSigma A2169) or 3TC (MilliporeSigma L1295) respectively. For Polθi-treated samples, cells were treated for 24 h with 10 μM of RP6685 (MedChemExpress, HY-151462).

### siRNA transfection

$0.5 \times 10^6$ cells were reverse transfected using RNAi max (Invitrogen) according to the manufacturer's instructions with 30 pmol of siRNAs of the indicated genes (Dharmacon, Supplementary Data 1.3) or scrambled nontarget siRNA (Dharmacon, Supplementary Data 1.3).

### CRISPR/Cas9 Ribonucleoprotein preparation

The crRNA (IDT) and the tracrRNA (IDT) were each resuspended to a final concentration of 100 mM in IDT buffer and mixed in an equimolar solution to a final concentration of 50 mM, heated at 95 °C for five minutes, and cooled to room temperature to form the crRNA:tracrRNA duplex. For each sample, 100 pmol of crRNA:tracrRNA duplex and 100 pmol of Cas9 enzyme (IDT) were diluted in PBS1X and incubated for 20 min at room temperature for ribonucleoprotein formation.

### Lentiviral production

For each transfection reaction, 5 μg of RRE, 3 μg of VSVG, and 2.5 μg of REV plasmid DNA were mixed with 20.5 μg of BFP plasmid DNA, 62 μg/mL polyethylenimine, and 150 mM sodium chloride. The reaction was incubated at room temperature for 15 min and then added to a plate of 10 cm HEK293T cells at ~70–80% confluence. Cells were incubated at 37 °C overnight. Fresh media was replaced and the cells were left to recover for 6–8 h before collecting the first viral supernatant. Fresh media was replaced, and after an additional 24 h, the second viral supernatant was collected. This was repeated for the collection of a third viral supernatant.

### BFP-to-GFP conversion assay

To deliver Cas9 and the gRNA against the BFP sequence (Supplementary Data 1.5), a CRISPR/Cas9 ribonucleoprotein was formed. $10^6$

cells were collected, washed with PBS1X, and resuspended in 100 µl of SF nucleofection buffer (Lonza). 5 µl of the ribonucleoprotein mixture was added to the cells, as well as 1 µl of the repair donor (100 µM). Reaction mixtures were electroporated in 4D Nucleocuvettes (Lonza) with the DS-150 program, incubated in the cuvette at 37 °C for 8 min with RPMI media, and transferred to culture dishes containing pre-warmed media. Cells were incubated for 72 h and then analyzed for blue or green fluorescence via flow cytometry.

### Genomic DNA extraction and PCR amplification for BFP-to-GFP conversion Assay

To genotype clones from 96-well plates, cells were resuspended in gDNA "dirty" lysis buffer supplemented with 10 mg/ml of Proteinase K. Cells were incubated overnight at 55 °C, and Proteinase K was inactivated by incubating the plate for five minutes at 65 °C. This DNA (~ 5 µl) was amplified via PCR with primers spanning the deletion site. The thermocycler was set for one denaturing cycle at 95 °C for three minutes, 35 denaturing cycles at 95 °C for 15 s, annealing at 60 °C for 15 s, extension at 68 °C for 40 s, and one final extension cycle at 68 °C for five minutes before being held at 12 °C.

### Native PAGE

To check the purity of the chimera donors purchased from IDT, we run them on Native Polyacrylamide Gel Electrophoresis (Native PAGE) in the presence or absence of RNAse A. The separating gels were prepared at 6% from acrylamide and bis-acrylamide solutions 29:1 in TBE. Gels were pre-run at 160 V for one hour before loading the samples. Before loading, the chimera donors were treated with 10 µg of RNaseA for 30 min at 37 °C. Gels were run at 120 V until the ladder reached the end. Gels were stained with Ethidium bromide for 15 min and then washed three times in H2O. GE Typhoon FLA 9000 Gel Scanner was used to detect the signal.

### Western blotting analysis

Cells were collected by trypsinization and lysed in RIPA buffer (25 mM Tris-HCl pH 7.6, 150 mM NaCl, 0.1% SDS, 1% NP-40, 1% sodium deoxycholate). After two cycles of water-bath sonication at medium settings, lysates were incubated at 4 °C on a rotator for an additional 30 min. Lysates were clarified by centrifugation for 30 min at $16,100 \times g$ at 4 °C, and the supernatant was quantified using the enhanced BCA protocol (Thermo Fisher Scientific, Pierce). Equivalent amounts of proteins were separated by SDS−PAGE and transferred to a nitrocellulose membrane. Membranes were blocked in 5% milk in TBST (137 mM NaCl, 2.7 mM KCl, 19 mM Tris-Base, and 0.1% Tween-20) or 5% BSA in TBST in the case of phosphorylated proteins for at least one hour at room temperature. Incubation with primary antibodies was performed overnight at 4 °C. Membranes were washed and incubated with HRP-conjugated secondary antibodies, developed with Clarity ECL (Bio-Rad), and acquired with a ChemiDoc MP Imaging System (Bio-Rad) and ImageLab v.5.2 (Bio-Rad). γ-tubulin, Lamin B, GAPDH, and vinculin were used as loading controls. The primary antibodies included FLAG (Clone M2, Sigma; 1:10000 dilution), 53BP1 (NB100-304, Novus Biologicals; 1:1000 dilution), HNRNPK (sc-28380, Santa Cruz; 1:1000 dilution), HNRNPC (sc-32308, Santa Cruz; 1:1000 dilution), GAPDH (0411, Santa Cruz, 1:10000 dilution), Vinculin (13901, Cell Signaling; 1:1000 dilution), and γ-tubulin (GTU-88; Sigma Aldrich; 1:5000 dilution). The secondary antibodies were mouse IgG HRP-linked (NA931, GE Healthcare; 1:5000) or rabbit IgG HRP-linked (NA934, GE Healthcare; 1:5000).

### qPCR validation of gene expression RT−qPCR

Total RNA was purified with the NucleoSpin RNA Clean-up (Macherey-Nagel) following the manufacturer's instructions. Genomic DNA was eliminated by on-column digestion with DNase I. A total of 1 µg of RNA was reverse transcribed using iScript Reverse Transcription Supermix

(Bio-Rad), and cDNA was diluted 1:10. Reactions were run with ssoAdvanced SYBR Green Supermix (Bio-Rad) with standard cycling conditions. Relative gene expression was normalized using *ACTB* as a housekeeping gene, and all calculations were performed using the ΔΔCt method. qPCR Primers are listed below in Supplementary Data 1.4.

### AAVS1-seq assay

$0.15−0.25 \times 10^6$ cells/well were seeded in six-well plates and treated with the respective siRNA as described above. Forty-eight hours post-knock-down, cells were transfected with 2 µg of CRISPR plasmid (pX300) directed to the *AAVS1* locus along with 10 µl of the 10 µM donor oligo (see Supplementary Data 1.2 for details) using Lipofectamine 3000 (Invitrogen). AAVS1 T2 CRISPR in pX330 was a gift from Masato Kanemaki (Addgene plasmid # 72833)[68]. Twenty-four hours following transfection, cells were harvested and gDNA was extracted using the DNeasy Blood & Tissue Kit (Qiagen). To measure the use of transcript RNA as a template, the pMJ1.19 plasmid transcribing a donor RNA complementary to the *AAVS1* locus was used.

### DNA library preparation, HiSeq sequencing for AAVS1-seq

Initial DNA sample quality assessment, library preparation, and sequencing were conducted at Azenta (South Plainfield, NJ, USA). Genomic DNA samples were quantified using a Qubit 2.0 Fluorometer (Life Technologies, Carlsbad, CA, USA). Locus-specific primers (oMJ80 and oMJ81, Supplementary Data 1.6) were used to amplify target sequences. PCR products were cleaned up, and sequencing libraries were prepared using the NEBNext Ultra DNA Library Prep Kit according to the manufacturer's protocol. In brief, amplicons were end-repaired and adenylated at the 3'ends. Adapters were ligated to the DNA fragments, and adapter-ligated DNA fragments were enriched and indexed with limited-cycle PCR. The adaptor-ligated sequencing libraries were validated on the Agilent TapeStation (Agilent Technologies, Palo Alto, CA, USA) and quantified by using Qubit 2.0 Fluorometer (Invitrogen, Carlsbad, CA) as well as by quantitative PCR (KAPA Biosystems, Wilmington, MA, USA). DNA libraries were multiplexed in equal molar mass and loaded on an Illumina HiSeq instrument according to the manufacturer's instructions (Illumina, San Diego, CA, USA). Sequencing was performed using a $2 \times 150$ paired-end (PE) configuration; the HiSeq Control Software conducts image analysis and base calling on the HiSeq instrument. Illumina Reagent/kits for DNA library sequencing cluster generation and sequencing were used for enriched DNA sequencing.

Paired-end sequencing data were analyzed using CRISPRResso2, which aligns reads to the target region using a global alignment algorithm after merging read pairs with FLASh[32]. Each unique mutational signature was identified utilizing a quantification window of 25 base pairs around the cut site (for sgRNA sequence, see Supplementary Data 1.5). For each sample, allele fractions for these events were calculated by counting the number of reads with respective mutational signatures identified by CRISPRResso2 and dividing the count by the total reads. The fraction of indel reads was calculated by dividing the read count for mutational signatures by the number of reads harboring indels in the sample.

### Droplet digital PCR (ddPCR)

Custom assays specific for detecting mutations in AAVS1 were ordered through Bio-Rad. Primers and probes for ddPCR are listed in Supplementary Data 1.1. Cycling conditions were tested to ensure optimal annealing/extension temperature and optimal separation of positive from empty droplets.

After PicoGreen quantification, 9–27 ng gDNA generated from the AAVS1-seq assay was combined with locus-specific primers, FAM- and HEX-labeled probes, BamHI, and digital PCR Supermix for probes (no dUTP). All reactions were performed on a QX200 ddPCR system (Bio-

Rad catalog # 1864001), and each sample was evaluated in technical replication of 2–8 wells. Reactions were partitioned into a median of ~14 K droplets per well using the QX200 droplet generator. Emulsified PCRs were run on a 96-well thermal cycler using cycling conditions identified during the optimization step (95 °C 10'; 40 cycles of 94 °C 30' and 60 °C 1'; 98 °C 10'; 4 °C hold). Plates were read and analyzed with the QuantaSoft software to assess the number of droplets positive for each sample.

## Validation of the RNA-transcript-PCR/gel/sequencing

Total RNA was purified using the Quick-DNA/RNA™ Miniprep Plus (Zymo Research) following the manufacturer's instructions. Genomic DNA was eliminated by on-column digestion with DNase I. A total of 2 μg of RNA was reverse transcribed using SuperScript™ IV VILO (Invitrogen). cDNA extracted from HEK293T cells transfected with the pMJ1.19 plasmid was amplified via PCR with either Primer Pair 1 (oMJ38-oMJ60) or Primer Pair 2 (oMJ39-oMJ61) (Supplementary Data 1.6) using Q5 master mix (NEB) under the following conditions: samples were denatured for 1 min at 98 °C for one cycle followed by 18 cycles of 98 °C for 10 s, 55 °C for 30 s, and 72 °C for 20 s, the final extension step was performed for one cycle at 72 °C for 2 min. PCR samples were then run on a 1% Tris-acetate EDTA agarose gel and visualized using the Bio-Rad Chemidoc XRS system. The amplicons were confirmed using Sanger sequencing performed by Azenta (South Plainfield, NJ, USA).

## Targeted CRISPR/Cas9 screens

CRISPR screens were performed as previously described[69]. HEK293T cells were transduced with a lentiviral DDR library at a low MOI (~ 0.2–0.3) and selected with 4 μg/ml of puromycin for 48 h post-transduction, which was considered the initial time point (day 0). Cells were grown for 10 days and then divided into three subpopulations. One population was kept in culture for an additional 4 days and was considered the non-treated sample. The other two populations were subjected to nucleofection with CRISPR/Cas9 at the BFP locus and either the DNA$^{GFP}$ or DNA/RNA$^{6R}$ donor at T10 and FACS sorted for GFP$^+$ vs BFP$^-$GFP$^-$ populations at T14. Sample cell pellets were frozen at each time point for genomic DNA (gDNA) isolation. A library coverage of ≥500 cells per sgRNA was maintained at every step. gDNA from cell pellets was isolated using Midi Kit (ZymoResearch) and genome-integrated sgRNA sequences were amplified by PCR using the Q5 Mastermix (New England Biolabs Next UltraII). i5 and i7 multiplexing barcodes were added in a second round of PCR, and final products were sequenced on Illumina HiSeq2500 or NextSeq500 systems to determine sgRNA representation in each sample. MAGeCK was used to identify essential genes[41] and top hits.

## Tumor-data analysis (whole intron deletion identification)

Mutation data from the MSK-IMPACT solid tumor cohort (64,544 samples, 56,322 patients) was systematically scanned using a script to identify WIDs[26,27]. Canonical intron-exon boundaries were obtained from Ensembl transcript files (GRCh37). For the 73,030 deletions in the MSK-IMPACT cohort, intron-exon boundaries in the reference genome were compared with deletion boundaries to identify whole intron deletions. To account for alignment discrepancies, margins of +/− 2 bp were allowed between intron boundaries and deletion boundaries on both edges (Supplementary Data 3.1).

Of the PCAWG cohort (1902 patients and samples), we identified whole intron deletions from 122,712 deletions longer than 10 base pairs using the same approach as mentioned above (Supplementary Data 3.3).

## RNA extraction from tumor samples

Following Institutional Review Board (IRB) approval, formalin-fixed paraffin-embedded (FFPE) tissues of 10 cases were retrieved from the pathology archives of Memorial Sloan Kettering Cancer Center (MSK).

Two pathologists (F.P. and T.V.) reviewed cases and included tumors arising from different anatomic locations (Supplementary Data 3.2). Cases were microdissected from ten eight-micron-thick histologic sections under a stereomicroscope (Olympus SZ61) to ensure a tumor content ≥80%. RNA was extracted using the RNAeasy FFPE kit (Qiagen) and subjected to RNA-sequencing at MSK Integrated Genomics Operation (IGO).

## RNA-seq on tumor samples

After RiboGreen quantification and quality control by Agilent BioAnalyzer, 0.5–1 μg of total RNA with DV200 percentages varying from 17 to 38% underwent ribosomal depletion and library preparation using the TruSeq Stranded Total RNA LT Kit (Illumina catalog # RS-122-1202) according to instructions provided by the manufacturer with 8 cycles of PCR. Samples were barcoded and run on a NovaSeq 6000 in a PE150 run, using the NovaSeq 6000 S4 Reagent Kit (300 Cycles) (Illumina).

## RNA-seq analysis

RNA sequencing reads were first examined using FASTQC[70], then Illumina universal adapters were trimmed by cutadapt[71]. The trimmed reads were aligned to the GRCh37 human genome using STAR RNA-Seq aligner[72], and then mapped single-end reads from transcripts were counted using the GenomicAlignments package in Bioconductor[73,74]. Read counts were further transformed into transcripts per million normalized for gene length.

## Whole intron deletion PCR confirmation

Patient DNA samples were processed and procured from the MSKCC IGO core facility. Genomic DNA was amplified using primers, as mentioned in Supplementary Data 1.7 and Q5 master mix (NEB), under the following conditions. Samples were denatured for 3 min at 98 °C for one cycle followed by 28 cycles of 98 °C for 10 s, 60 °C (*GNAS*) or 65 °C (*HLA*) for 30 s, and 72 °C for 20 s, the final extension step was performed for one cycle at 72 °C for 2 min. PCR samples were then run on a 1% Tris-acetate EDTA agarose gel and visualized using the Bio-Rad Chemidoc XRS system. The amplicons were confirmed using Sanger sequencing performed by Azenta (South Plainfield, NJ, USA). The corresponding patients tested from the MSK-IMPACT cohort were: P1-IMPACT_WID_38, P2- IMPACT_WID_12, P3- IMPACT_WID_29, P4-IMPACT_WID_34.

## Mathematical modeling

We developed a simulation strategy to quantify the likelihood of observing WIDs by random chance, rather than through any specific mechanism. This strategy simulates a cohort of deletions based on MSK-IMPACT data, taking into account both the genomic locations of mutations and the length of the deletion. We investigate whether the observed occurrence of WID in MSK-IMPACT data would exceed chance expectations based on simulated MSK-like cohorts.

The simulation approach learns the probability distribution of deletion lengths from the actual MSK-IMPACT data and uses this distribution as the probability to assign lengths to the simulated deletions. MSK-IMPACT is a targeted panel with only certain genomic regions being sequenced; here, we used these regions to reflect the space where simulated deletions could occur. Each interval within the MSK-IMPACT panel varies in its observed abundance of detected deletions in the actual MSK-IMPACT data, likely depending on interval length (some intervals are longer, and this could also increase the likelihood of deletions occurring) or even on biological reasons. To take this into account, we also used the probability distribution of deletions from the actual MSK-IMPACT data to assign probabilities to specific intervals in the simulation.

Each simulated cohort contains 73,030 deletions, mirroring the characteristics of the MSK-IMPACT cohort. Simulating a deletion involves three steps: (1) Randomly selecting a panel interval based on

the observed probability distribution. (2) Randomly determining the starting position within the selected interval. (3) Randomly choose the deletion length according to the observed probability distribution. The end position of each simulated deletion is defined as the starting position plus the deletion length. Deletions must adhere to two constraints: staying within the same interval or ending at the start of the next interval to be detected by the MSK-IMPACT panel.

Each deletion was annotated after constructing a simulated dataset reflecting the MSK-IMPACT cohort, including simulated whole intron deletions. It should be noted that analyses based on the simulation strategy may be influenced by inherent randomness, leading to fluctuating results based on the random seed used. To address this, we create a cohort of 10,000 MSK-IMPACT-like deletion cohorts, employing different seed numbers for each cohort, ensuring distinct sets of simulated deletions. The *p*-value was calculated by determining the frequency at which a simulated cohort exhibits an equal or more significant number of WID compared to the actual MSK-IMPACT data.

Similarly, we generated 10,000 deletion cohorts resembling PCAWG data. Given that PCAWG samples were subjected to whole-genome sequencing (WGS), modifications to the simulation strategy were introduced. We focused the simulation on the subset of PCAWG deletions occurring within genes, as deletions in intergenic regions are irrelevant to this analysis. Within genes, the frequency of deletions based on the simulations was compared to the observed probability distribution in the actual PCAWG data. This approach sought to capture the relative abundance of deletions within genes, considering factors such as gene length.

### Intron loss-seq assay

To deliver Cas9 and the gRNA against *CALR* intron 2, *GNAS* intron 11, and *IL3* intron 4 (Supplementary Data 1.5), a CRISPR/Cas9 ribonucleoprotein complex was formed. $10^6$ cells were collected, washed with PBS, and resuspended in 100 µl of nucleofection buffer. Five microlitres of the ribonucleoprotein mixture was added to cells with SF buffer and either 1 µl (100 µM) of the repair donor (120 bp ssDNA oligo chimera with 6 ribonucleotides spanning the exon-exon junction sequence (DNA$^{CALR-6R}$) or no donor. Cells were electroporated using the 4D Nucleofector (Lonza) with the cell line specific program (DS-150). Cells were incubated at 37 °C for 8 min with RPMI media and then transferred to culture dishes containing pre-warmed media in which they were incubated for 72 h. Genomic DNA was extracted using the Quick DNA Miniprep Kit (ZymoResearch) and then amplified by PCR with primers at the flanking exons (Supplementary Data 1.7) using Q5 Mastermix (New England Biolabs Next UltraII). i5 and i7 multiplexing barcodes were added in a second round of PCR, and final products were sequenced on Illumina NovaSeq X by the MSK IGO sequencing core using PE150 sequencing. CRISPResso2 was used to map reads to either the reference amplicon or the amplicon with a perfect intron deletion[32]. A quantification window of 3 bp on either side of the exon-exon junction site was used to label and filter out imperfect intron loss as reads mapped to intron loss but containing insertions or deletions within this 6 bp window. The percent of intron loss reads was calculated by dividing the read count for perfect intron loss by the total number of reads (reference + perfect intron loss + imperfect intron loss). For siRNA-mediated knockdown experiments, $10^6$ cells were seeded in 6 cm plates and treated with the respective siRNA as described above. Forty-eight hours post-knock-down, cells were collected for nucleofection with CRISPR/Cas9 ribonucleoprotein. For AZT-treated samples, cells were treated for 24 h with 10 µM/mL of AZT (MilliporeSigma A2169).

### Statistics

All statistical analysis was performed with GraphPad Prism 9. Sample sizes and the statistical tests used are specified in the figure legends.

### Reporting summary

Further information on research design is available in the Nature Portfolio Reporting Summary linked to this article.

## Data availability

Raw sequencing data for AAVS1-seq has been deposited in NCBI's Sequence Read Archive (SRA) under BioProject Accession Number PRJNA1236828. The RNA-seq data generated in this study have been deposited to NCBI's Gene Expression Omnibus with the GEO Series Accession Number GSE291098. Data generated by the Intron-loss assay is deposed through GEO Series Accession Number GSE290535. The raw sequencing data for the MSK-IMPACT and PCAWG analysis is protected and cannot be broadly available due to privacy laws; patient consent to deposit raw sequencing data was not obtained. De-identified data are available under restricted access to protect patient privacy per federal and state law. Raw data may be requested from sfeira@mskcc.org and powells@mskcc.org with appropriate institutional approvals with responses expected within four weeks. Tumor DNA and RNA from MSK-IMPACT samples were collected under the IRB 12-245 protocol. Requests for materials and/or questions regarding any of the constructs, cell lines, or other data described should be addressed to the corresponding authors. Source data is available in Mendeley https://doi.org/10.17632/zsvrjcwz5s.1. Source data are provided in this paper Source data are provided with this paper.

## Code availability

All script used to analyze Whole Intron Deletions (WIDs) can be obtained from the GitHub repository at https://github.com/Yingjie848/project-RNA-templated-DSB-repair-public.

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

## Acknowledgements

We acknowledge using the Integrated Genomics Operation Core, funded by the NCI Cancer Center Support Grant (CCSG, P30 CA08748), Cycle for Survival, and the Marie-Josée and Henry R. Kravis Center for Molecular Oncology. Illustrations were created with *in BioRender. Jalan, M. (2025)* https://BioRender.com/9tmc1xb. We thank Erik Anderson for help with designing the RNA-donor plasmid. We thank Ronglai Shen for the helpful discussions on developing the simulations. We thank Martin Stojaspal for technical support with the native PAGE. We thank Raj Chari and Genome Modification Core of NCI for assistance with synthesizing the DDR-focused CRISPR/Cas9 library. We acknowledge the Powell and Sfeir lab members for commenting on the manuscript. This work is supported by grants from NIH/NCI (R01CA229161 and U01CA231019) for A.S., M.J. is supported by the AACR-Swim Across America Cancer Research Fellowship (20-40-64-JALA). S.N.P. is funded by the Breast Cancer Research Foundation and NIH/NCI P50 CA247749. J.S.R.-F. is funded in part by the Breast Cancer Research Foundation, Susan G Komen Leadership grant, and the NIH/NCI P50 CA247749 grant. P.C.B. is supported by the NCI awards P30CA016042, R01CA244729, and U2CCA271894. H.S. is a Marie-Josée Kravis Women in Science Endeavor Graduate Student Fellow.

## Author contributions

M.J., A.B., S.N.P., and A.S. conceived the experimental design and implemented the study. A.B. performed the BFP-to-GFP assay with help from H.S. and J.W., and M.J. performed the AAVS1-seq assay with help from N.M.D. and J.G-A. and K.S.A. A.D. performed the CRISPR screen with A.B., and H.S. helped with the screen analysis. S.A.-S., S.H., and D.H. helped with the AAVS1-seq assay optimization. M.J., N.M.D., J.P., Y.Z., A.G., T.N.Y., P.C.B., N.R. and J.S.R.-F. contributed to the computational analysis; J.P., Y.Z., and N.R. analyzed the CRISPResso data; J.P. developed the computational pipeline to search for tumor-specific WIDs with help from Y.Z., T.N.Y., P.S., and A.N-G., while A.G. developed the mathematical modeling. Y.Z., F.P., T.V., and E.M.D. helped with the sample collection and analysis for the RNA-seq data. H.S. optimized and performed the intron-loss assay in cells. B.A. generated the DDR-focused CRISPR/Cas9 library. The manuscript was prepared by M.J. and A.B. and revised by N.M.D., H.S., S.N.P., and A.S. with input from all authors.

## Competing interests

A.S. is a co-founder, consultant, and shareholder for REPARE Therapeutics. S.N.P is a consultant for AstraZeneca, Varian Medical Systems, and Philips. J.S.R.-F. reports current employment at AstraZeneca and stocks in AstraZeneca, Repare Therapeutics, Paige.AI; J.S.R.-F. previously held a fiduciary role in Grupo Oncoclinicas and consulted with Goldman Sachs Merchant Banking, Bain Capital, Repare Therapeutics, Paige.AI, Volition Rx and MultiplexDx. P.C.B. sits on the Scientific Advisory Boards of Intersect Diagnostics Inc., BioSymetrics Inc., and Sage Bionetworks. The remaining authors declare no competing interests.
