## [Transparent Peer Review file · Nature Communications]

RNA Transcripts Serve as a Template for Double-Strand Break Repair in Human Cells

Corresponding Author: Dr Agnel Sfeir

Version 0:

Reviewer comments:

Reviewer #1

(Remarks to the Author)

In this revised manuscript, the authors find a possible role for RNA-templated DSB repair in generating whole-intron deletions (WID) mediated by reverse transcriptase activity by Pol zeta. The authors have already addressed many issues raised and have strengthened their manuscript considerably. Also the figures have been much revised which helps to convey the results much clearer. There are two points that I still want to raise:

1. The authors have now included an important assay where they test if WID is mediated by REV3L (Figure 5B). What is surprising to me is that the fraction of WIDs is only reduced by about 40%. That difference is much more profound with a DNA/RNA chimera (i.e. Fig. 4B, 4G). Since the knockdown conditions are very similar between those experiments, and thus can be excluded as an argument for the difference found, am I correct to conclude that there is at least an REV3-independent way of performing WID? It would for example have helped to combine the sgCALR experiment with an REV3 knockdown, although I am reluctant to propose additional experiments. Perhaps the authors can address this textually? Perhaps LINE1 or other reverse transcriptases contribute more to WID than to DNA/RNA chimeras?

2. In the final part of the discussion the authors suggest that RT-DSBR may play a role in non-dividing cells that lack HR. To me this suggestion directly contradicts the findings in the CRISPR screen where HR and resection factors were shown to be required for editing via ssDNA and DNA/RNA6R (i.e. BRCA1, HELQ, NBN, EXO1). While it is an attractive thought to think of RT-DSBR as an ultimate attempt to restore gene function in the case of a DSB it seems to be not supported by the author's own data.

Minor points:

1. Figure 2C – Please realign the dots such that they are aligned with the ticks and distributed evenly vertically. That also applies to the POLQ labels in the same figure.
2. Figure S1D – ‘Scrembel’ -> ‘Scramble’

Reviewer #2

(Remarks to the Author)

This manuscript is acceptable for publication in Nature Communications.

REVIEWERS' COMMENTS

Reviewer #1 (Remarks to the Author):

In this revised manuscript, the authors find a possible role for RNA-templated DSB repair in generating whole-intron deletions (WID) mediated by reverse transcriptase activity by Pol zeta. The authors have already addressed many issues raised and have strengthened their manuscript considerably. Also the figures have been much revised which helps to convey the results much clearer. There are two points that I still want to raise:

1. The authors have now included an important assay where they test if WID is mediated by REV3L (Figure 5B). What is surprising to me is that the fraction of WIDs is only reduced by about 40%. That difference is much more profound with a DNA/RNA chimera (i.e. Fig. 4B, 4G). Since the knockdown conditions are very similar between those experiments, and thus can be excluded as an argument for the difference found, am I correct to conclude that there is at least an REV3-independent way of performing WID? It would for example have helped to combine the sgCALR experiment with an REV3 knockdown, although I am reluctant to propose additional experiments. Perhaps the authors can address this textually? Perhaps LINE1 or other reverse transcriptases contribute more to WID than to DNA/RNA chimeras?

As we had already done the intron-loss assay in cells treated with LINE inhibitor, we now include the results that conclude that L1 reverse transcriptase also does not affect RT_DSBR in that setting. The data is included in the final version of the manuscript (supplemental figure 6)

2. In the final part of the discussion the authors suggest that RT-DSBR may play a role in non-dividing cells that lack HR. To me this suggestion directly contradicts the findings in the CRISPR screen where HR and resection factors were shown to be required for editing via ssDNA and DNA/RNA6R (i.e. BRCA1, HELQ, NBN, EXO1). While it is an attractive thought to think of RT-DSBR as an ultimate attempt to restore gene function in the case of a DSB it seems to be not supported by the author's own data.

We thank the reviewer for sharing their perspective. We believe it is appropriate for us – as authors of this manuscript – to include a contextual discussion of our findings and to speculate on their potential implications, especially in the discussion section. While we understand the reviewer's concern, we respectfully disagree. RT-DSBR may still be relevant in non-dividing cells such as neurons, where the genetic requirements for repair differ from those in dividing cells, as we state.

Minor points:

1. Figure 2C – Please realign the dots such that they are aligned with the ticks and distributed evenly vertically. That also applies to the POLQ labels in the same figure.

The alignment for all panels have been fixed.

2. Figure S1D – ‘Scremble’ -> ‘Scramble’

This has been fixed.

Reviewer #3 (Remarks to the Author):

This manuscript is acceptable for publication in Nature Communications.